# Exploring the Use of Fruit Callus Culture as a Model System to Study Color Development and Cell Wall Remodeling during Strawberry Fruit Ripening

**DOI:** 10.3390/plants9070805

**Published:** 2020-06-27

**Authors:** Pablo Ric-Varas, Marta Barceló, Juan A. Rivera, Sergio Cerezo, Antonio J. Matas, Julia Schückel, J. Paul Knox, Sara Posé, Fernando Pliego-Alfaro, José A. Mercado

**Affiliations:** 1Instituto de Hortofruticultura Subtropical y Mediterránea “La Mayora” (IHSM-UMA-CSIC), Departamento de Botánica y Fisiología Vegetal, Universidad de Málaga, 29071 Málaga, Spain; pabloric@uma.es (P.R.-V.); juan89_uma@hotmail.es (J.A.R.); scerezo@uma.es (S.C.); antoniojmatas@uma.es (A.J.M.); sarapose@uma.es (S.P.); ferpliego@uma.es (F.P.-A.); 2IFAPA Centro de Málaga, Cortijo de la Cruz s/n, 29140 Málaga, Spain; marta.barcelo@juntadeandalucia.es; 3Department of Plant and Environmental Sciences, University of Copenhagen, 1871 Frederiksberg, Denmark; js@glycospot.dk; 4Centre for Plant Sciences, Faculty of Biological Sciences, University of Leeds, Leeds LS2 9JT, UK; j.p.knox@leeds.ac.uk

**Keywords:** carbohydrate microarray, callus culture, cell culture, cell wall, extensin, fruit ripening, fruit softening, pectin

## Abstract

Cell cultures derived from strawberry fruit at different developmental stages have been obtained to evaluate their potential use to study different aspects of strawberry ripening. Callus from leaf and cortical tissue of unripe-green, white, and mature-red strawberry fruits were induced in a medium supplemented with 11.3 µM 2,4-dichlorophenoxyacetic acid (2,4-D) under darkness. The transfer of the established callus from darkness to light induced the production of anthocyanin. The replacement of 2,4-D by abscisic acid (ABA) noticeably increased anthocyanin accumulation in green-fruit callus. Cell walls were isolated from the different fruit cell lines and from fruit receptacles at equivalent developmental stages and sequentially fractionated to obtain fractions enriched in soluble pectins, ester bound pectins, xyloglucans (XG), and matrix glycans tightly associated with cellulose microfibrils. These fractions were analyzed by cell wall carbohydrate microarrays. In fruit receptacle samples, pectins were abundant in all fractions, including those enriched in matrix glycans. The amount of pectin increased from green to white stage, and later these carbohydrates were solubilized in red fruit. Apparently, XG content was similar in white and red fruit, but the proportion of galactosylated XG increased in red fruit. Cell wall fractions from callus cultures were enriched in extensin and displayed a minor amount of pectins. Stronger signals of extensin Abs were detected in sodium carbonate fraction, suggesting that these proteins could be linked to pectins. Overall, the results obtained suggest that fruit cell lines could be used to analyze hormonal regulation of color development in strawberry but that the cell wall remodeling process associated with fruit softening might be masked by the high presence of extensin in callus cultures.

## 1. Introduction

Ripening of fleshy fruits is a complex developmental process involving changes in color, flavor, and texture that make the tissue edible to seed-dispersing animals [1]. From an agricultural point of view, textural changes are of major importance since the loss of firm texture is the main determinant of the postharvest shelf life in most commodities [2,3]. Furthermore, fruit texture is one of the main attributes for the acceptance in the market from the consumer’s point of view [2]. Soft fruits such as strawberry acquire an undesirable melting texture very soon after ripening, increasing their susceptibility to pathogen attack and reducing their shelf life to a few days [4,5].

It is generally accepted that the modification of the mechanical properties of the primary cell walls due to cell wall disassembly, the reduction of intercellular adhesion as a result of middle lamella dissolution, and the reduction in cell turgor are the major causes of fruit softening [6,7,8,9]. Among these three factors, most studies have focused on the cell wall disassembly process taking place during fruit ripening; however, despite the large amount of information available, a general model of cell wall remodeling leading to fruit softening remains elusive [9]. Functional analyses of genes encoding pectinases such as polygalacturonase and pectate lyase point to the pectin fraction as a key factor involved in strawberry softening [10,11].

An additional area of active research in strawberry is the genetic regulation of fruit color. Anthocyanin is the principal pigment leading to the red color of ripe strawberry fruit [12,13]. Some key genes involved in the biosynthesis and the regulation of flavonoid/phenylpropanoid compounds have been characterized using transgenic approaches [12,14,15,16]. All these studies have been performed with whole strawberry fruit, a complex organ containing different tissues that differ in their metabolism and that undergo ripening at different rates.

In many cases, cell cultures can be used as simple model systems to study developmental processes, such as stem cell regulation, mineral deficiency, or disease and stress responses [17,18,19,20]. In vitro cultures provide a source of uniform plant material that can be easy to handle, avoiding complex interactions among different plant organs and/or tissues. In the case of fruit, calyx and fruit cultures as well as callus derived from fruit tissues have been employed to study different aspects of ripening, such as hormonal regulation [21], flavor and color development [22], or defense responses [23]. In strawberry, callus from immature fruit has been used to analyze phenol metabolism during the in vitro culture period and its relationship with cell growth [24,25]. Strawberry leaf callus has also been employed to characterize anthocyanin synthesis [26,27]. As regards cell wall metabolism, the effect of plant growth regulators in cell wall composition of callus cultures obtained from immature apple fruits has been determined [28]. As far as is known, the remodeling of cell walls associated with fruit softening has not yet been addressed using callus cultures. The aims of this study were firstly to produce cell lines from strawberry fruits at different developmental stages and secondly to characterize these lines to determine if they could be a useful model system to gain insight into the fruit ripening process, particularly into the regulation of anthocyanin synthesis and the cell wall disassembly associated with fruit softening.

## 2. Results

### 2.1. Effect of Hormonal Balance and Explant Type on Strawberry Callus Development

To optimize the generation of callus from strawberry tissue, leaf explants from micropropagated plants were cultured in N_30_K medium supplemented with a combination of benzyladenine (BA) and 2,4-Dichlorophenoxyacetic acid (2,4-D) at different concentrations. The presence of 2,4-D in the medium was needed to induce the formation of friable callus; by contrast, BA diminished callus production. After 8 weeks of culture, the highest amount of callus was formed in leaf explants cultured in media only supplemented with 2,4-D (Figure 1). Calli induced in the different media were isolated and recultured into the same fresh medium. In all media only supplemented with 2,4-D as growth regulator, callus showed a good growth rate reaching mean weight values of 4.6, 4.7, and 5.3 g in 11.3, 22.6, and 45.2 µM 2,4-D medium, respectively, at the end of the experiment after 16 weeks of culture. However, calli obtained in the media supplemented with BA did not increase their fresh weight after subsequent recultures to fresh medium.

Sections of fruit receptacle at three developmental stages, immature-green, white, and ripe-red, were cultured in N_30_K medium supplemented with 11.3 µM 2,4-D to induce callus formation. Although the rate of explant contamination was high, especially in red fruit (>60%), several callus lines from fruit at each developmental stage could be recovered. One cell line was selected for each explant type and further multiplied (Figure 2). Afterwards, the growth curve of selected cell lines growing in N_30_K-11.3 µM 2,4-D medium was recorded (Figure 3). The highest growth rate was observed in callus from green-fruit followed by white- and red-fruit. Leaf callus displayed an intermediate behavior between white-fruit and red-fruit calli. The aspect of these cell lines was also different after 3 weeks of culture. White and red-fruit calli were white-beige with a creamy and watery texture. Green-fruit callus showed a similar color but a more compact and hard texture. Finally, leaf callus was yellowish and soft.

To further characterize the best conditions for strawberry callus proliferation, the effect of light incubation was analyzed. Thus, calli were cultured in N_30_K-11.3 µM 2,4-D and maintained in darkness or under a 16 h photoperiod of 40 µmol·m^−2^·s^−1^. Calli cultured under illumination exhibited a higher fresh weight than those incubated under darkness, with significant differences in the case of calli from green and white fruits (Figure 4). Interestingly, some sectors of calli incubated in light developed red pigmentation after three weeks of culture, especially in the case of leaf and red-fruit cell lines (Figure 2). Average value of anthocyanin content, measured as pelargonidin-3-glucoside, was 0.24 ± 0.03 mg/100 g fresh weight in calli cultured in darkness; when calli were grown in light, the amount of anthocyanin increased, and the mean values ranged between 0.34 ± 0.07 and 0.45 ± 0.09 mg/100 g in green-fruit and leaf callus, respectively.

### 2.2. Effect of ABA in the Production of Anthocyanin

To determine if abscisic acid (ABA) enhanced the production of anthocyanin in strawberry calli cultured in the presence of light, cell lines were incubated in N_30_K medium supplemented with different concentrations of ABA in the absence of auxin. A control treatment of callus cultured without growth regulators was also included. As previously observed, the transfer of calli cultured in the standard growth medium (11.3 µM 2,4-D) from dark to light conditions induced the production of anthocyanin, especially in leaf and red-fruit cell lines (Figure 5). The substitution of auxin by ABA at the lowest concentration (1 µM) notably increased anthocyanin production in calli from green fruits and to a lesser extent in red-fruit line (Figure 2 and Figure 5); however, leaf and white-fruit calli did not respond to this treatment. Higher ABA concentrations or the absence of growth regulators in the culture medium diminished cell viability, and most cells showed necrosis after 4 weeks of culture.

### 2.3. Analysis of Cell Wall Components by Comprehensive Microarray Polymer Profiling (CoMPP)

Cell walls from leaf and fruit cell cultures were extracted and treated with different solvents to yield fractions enriched in soluble and ester-bound pectins (water and sodium carbonate, respectively), xyloglucans (4M KOH), and matrix polysaccharides tightly linked into cellulose microfibrils (cadoxen). Cell walls from fruits at green, white, and red stages were subjected to the same fractionation process for comparison. Table 1 displays the antibodies used in the CoMPP analysis. A heatmap of the obtained carbohydrate microarray results is shown in Figure 6. LM27 mAb against grass xylan was included as negative control, and, as expected, no signal was detected in any sample. In the case of fruit cell walls, methylesterified homogalacturonan (HG), recognized by JIM7, was abundant in water fractions of green and white fruits, whereas red fruit showed higher signal of JIM5, which binds to low methylester and unesterified HG. Both HG and rhamnogalacturonan I (RGI) were detected in the sodium carbonate fraction as well as in KOH and cadoxen fractions, indicating that a large fraction of pectins remained linked to xyloglucan (XG) and/or entrapped in cellulose microfibrils. In general, the amount of extractable pectins increased from green to white fruits and then declined in red fruit. KOH and cadoxen fractions contained significant amounts of XG, recognized by LM15 and LM25 mAbs. As observed for pectin epitopes, the amount of extracted XG increased from green to white fruits; however, in red fruits, the LM15 signal for the XXXG motif of XG diminished, while the LM25 signal additionally recognizing XG galactosyl residues increased. Glucuronoxylan recognized by LM28 was also abundant in KOH and cadoxen fractions from green and red fruits. A low signal of arabinogalactan protein glycan (JIM13) was detected in the KOH fraction, increasing in amount as the fruit ripen.

Polysaccharide profiles of cell walls from callus cultures were quite different from the one described for fruits. In all cell wall fractions, HG and RGI pectin mAbs displayed a very low signal. The amounts of XG and xylan in KOH and cadoxen fractions were also lower than in fruit fractions. By contrast, the three mAbs against extensin, LM1, JIM11, and JIM20 showed strong signals in cell culture fractions and were highest in the sodium carbonate fraction. Extensin epitopes were not detected in fruit samples or in the water fraction from callus cultures. LM23 also showed a different profile in fruit and cell cultures. This mAb binds to xylogalacturonan, and it was detected in all fractions from callus lines except water fraction but was absent in fruit cell walls.

A principal component analysis (PCA) was performed to provide an overview of sample group separation (Figure 7 and Figure 8). In the four cell wall fractions, the first principal component accounted for most of the variability in the data set (>67%). Samples from fruit and cell cultures were separated along this first principal component, with fruit samples in the positive values and cell cultures in the negative values. The only exception was the water fraction from ripe fruit that appeared close to cell culture samples (Figure 7). mAbs against neutral carbohydrates and RGI backbone (LM5, LM6-M, INRA-RU1) as well as JIM7 against methylesterified HG correlated positively with the first dimension in the case of water fractions. In the sodium carbonate fraction, mAbs related to RGI and unesterified HG (LM19, LM18) contributed positively to the first principal component, while extensin mAbs (JIM11, JIM29, and LM1) contributed negatively. KOH and cadoxen fractions followed a similar trend, although XG related mAbs (LM23 and LM25) also correlated negatively with the first component. The results obtained indicate that the first PCA dimension, accounting for most of the variability in the samples, was related to the amount of pectins in the water fraction and the substitution of pectins by extensin epitopes in the rest of fractions.

The whole set of microarray data was subjected to hierarchical clustering (Figure 9; Appendix A). Samples were grouped in three main clusters (Figure 9). The cluster 1 comprised sodium carbonate, KOH, and cadoxen fractions from fruit cell cultures. This group was characterized for strong signals of extensin (JIM20, JIM11, LM1), xylogalacturonan (LM23) and mannans (LM21) mAbs, and low values of RGI (INRA-RU1, LM5, LM6-M, INRA-RU2) and HG (LM18, LM19). The cluster 2 included water fractions from both cell cultures and fruits and sodium carbonate fractions from fruits. This group was defined by the high signal of HG epitopes (JIM5 and JIM7) and the low values for XG (LM15 and LM25) and extensin mAbs. Finally, the third cluster comprised KOH and cadoxen fractions from fruits. This clade was characterized by the large abundance of XG, RGI, and HG epitopes and the low amount of extensin.

## 3. Discussion

### 3.1. Development of Callus Cultures from Fruit Receptacle

Callus cultures have been induced in strawberry from leaf [26], petioles [29], apical meristems [30], and immature fruit [25,31] using Murashige and Skoog (MS) [32] or Linsmaier and Skoog (LS) [33] medium supplemented with 2,4-D. Previous studies demonstrated that N_30_K, a mineral formulation with lower ionic strength than MS, was more suitable for micropropagation and regeneration of strawberry cv. ‘Chandler’ [34,35]. In this research, this medium was used successfully to obtain callus cultures in leaf explants from this genotype. The presence of 2,4-D was needed to induce callus formation, as previously observed in some strawberry genotypes [31]. The addition of BA to the induction medium diminished the amount of callus formed. Asahira and Kano [36] also found that callus was only established on a medium containing 2,4-D but lacking BA. N_30_K medium supplemented with a low 2,4-D concentration was also suitable for isolation and proliferation of cell cultures from fruit receptacle at different developmental stages. Explants from red fruit showed the lowest capacity to develop callus in this medium. Similarly, the proliferation rate of the cell lines obtained diminished accordingly to the developmental age of the initial explant, being higher in green immature fruits and lower in ripe fruits. As observed in this research, Hong et al. [31] reported that the rate of callus formation from strawberry fruit decreased with the age of the explant, but they were unable to obtain callus from nearly mature fruits collected four weeks after flowering. In other species, immature fruits have also been chosen as the preferred explant to initiate cell lines [37]; to the best of our knowledge, this is the first time that callus cultures have been obtained from highly differentiated mature fruit.

### 3.2. Anthocyanin Production in Strawberry Fruit Cultures

Light slightly increased the proliferation rate of green and white callus cultures, and it induced the accumulation of red pigmentation, especially in leaf and ripe-fruit lines. A similar observation was made in cv. ‘Brighton’ [31]. In strawberry fruit, anthocyanin biosynthesis is regulated by the transcription factor FaMYB10. This gene is mainly expressed in ripe receptacle and directly controls the expression of early and late-regulated biosynthesis genes involved in the flavonoid/phenylpropanid pathway, including anthocyanin production genes [38]. Kadomura-Ishikawa et al. [39] found that light and ABA independently regulate *FaMYB10* expression and therefore anthocyanin production in strawberry fruit, their effect being additive. The replacement of 2,4-D by ABA at low concentrations notably increased anthocyanin production in callus from immature green fruit but did not significantly affect that of the other lines. ABA and auxins play antagonistic effects on strawberry fruit ripening. Decreasing ABA biosynthesis in green fruit by virus induced gene silencing of *FaNCDE1*, a key gene in ABA biosynthesis, or injection of the ABA inhibitor fluridone impaired fruit development, obtaining a colorless fruit phenotype [40]. A similar phenotype was obtained when down-regulating ABA receptors [41]. Contrary to ABA, auxins produced by achenes reach a maximum level in green receptacle [42], and the removal of the achenes induced the accumulation of anthocyanin [43]. The results obtained in this research suggest large differences in the regulation of the flavonoid pathway among the different fruit cell lines. Auxin/ABA hormonal balance seems to be the main regulator of anthocyanin production in the cell line from green fruits; however, cultures from ripe fruits mainly respond to light, while those from white fruits showed a low capacity for anthocyanin production. The differential behavior of the fruit cell lines could be exploited to decipher the genetic regulation of pigment accumulation in strawberry fruit.

### 3.3. Characterization of Cell Walls from Cell Cultures

The pectin profile in cell wall fractions isolated from fruits showed that the amounts of HG and RGI epitopes increased from green to white fruit and then declined in red ripe fruit to very low levels. This pattern was observed in all fractions analyzed except in the water fraction from ripe fruit that contained significant amounts of low methyl ester HG recognized by JIM5. These findings support previous observations of pectin solubilization, i.e., a reduction in the amount of pectins tightly bound to the cell wall concomitant to an increase in water-soluble pectins as a typical feature of strawberry remodeling during fruit ripening [4,44,45,46]. It is noteworthy that a large fraction of pectins, especially RGI, was extracted with KOH and cadoxen. Pectin and xyloglucan can be covalently linked [47,48] and recently, Cornuault et al. [49] found that a sub-population of pectin was attached to XG in KOH fractions from different fruits, including strawberry. As regards hemicellulose, it has been reported that its content diminished during strawberry ripening in cultivars with contrasting fruit firmness [50,51]. Contrary to those results, the carbohydrate microarray showed that the signal intensity of extractable XG epitopes changed little during ripening of ‘Chandler’ fruit; however, the results obtained suggest a change in the chemical structure of XG in ripe fruit, increasing the amount of galactosylated XG recognized by LM25. Significant amounts of glucuronoxylans were also detected in KOH and cadoxen extracts and with an intriguing pattern of relatively lower abundance in white fruit. Heteroxylans are not abundant in primary cell walls of dicotyledonous plants, and the role of glucuronoxylan is unknown. Cornuault et al. [52] found glucuronoxylan associated with RGI-enriched fraction from potato tubers. However, our results indicate a negative correlation between the glucuronoxylan signal and RGI abundance in matrix glycan fractions.

Carbohydrate profiles from cell cultures were quite different from fruit. In callus cell walls, pectin antibodies showed a very weak signal. By contrast, callus cell wall extracts were enriched in extensin epitopes associated with tightly bound pectins (sodium carbonate fraction) and matrix glycans. Extensins are one of the main classes of hydroxyproline-rich glycoproteins (HRGPs) that contain multiple Ser-(Pro)3–5 repeats and Tyr motifs acting as cross-link sites [53,54]. These proteins are involved in the building and the maintenance of primary cell wall. During the cell plate formation, the self-assembling ability of extensins generates scaffolding networks that may serve as a template for pectin matrix assembly due to the acid-base interactions between extensin and pectin [53]. On the other hand, extensin cross-linking strengthens cell walls and many biotic and abiotic stresses induce extensin biosynthesis, e.g., pathogen attack or wounding [55,56,57]. In grape berries, extensin epitopes detected with LM1 and JIM20 increased at véraison when ripening started; however, its physiological function was unclear [58]. Contrary to grape, the CoMPP analysis showed that extensin was a minor component in the strawberry cell wall during fruit development. The high extensin content in the strawberry cell cultures may reflect a tightly cross-linked cell wall and could be related to the stressful conditions of in vitro tissue culture; in fact, high amounts of extensin have also been found in suspension cells from other species [59,60]. The stronger extensin epitope signals were detected in sodium carbonate fraction, especially in leaf and green fruit lines, suggesting that these proteins might be linked to pectins, as reported by [61]. This interaction could also explain the low label of mAb against pectins, i.e., pectin epitopes might be masked in pectin-extensin complexes, resulting in a low binding efficiency of HG and RGI antibodies. Indeed, pectin and other cell wall polysaccharides may be more tightly linked into the cell wall structures of callus cells and therefore less extractable to appear in the solubilized fractions. On the other hand, LM23 binding was detected in all cell wall fractions from callus samples except water, indicating the presence of xylogalacturonan (XGA) epitopes. This pectin has not previously been described in strawberry fruit, but it has been found in hairy regions of apple pectin, another member of Rosaceae family [62]. XGA has been related to cell adhesion in carrot-cultured cells [63].

Carbohydrate microarrays provide information about the relative levels of glycan epitopes but do not allow quantifying absolute levels of cell wall components [64]. In callus cultures, the large amounts of extensin and the apparent lower presence of pectin epitopes make it difficult to compare these cell walls with their corresponding samples from fruits using this approach. However, the hierarchical clustering of microarray data included in the same group the water fractions from both kinds of samples jointly with sodium carbonate fractions from fruits. It is therefore likely that pectin composition in these fractions were comparable. As previously indicated, polyuronides are extensively modified during strawberry ripening [4,44]. Other techniques such as carbohydrate structural analyses would be needed to precisely determine whether cell walls at different fruit developmental stages and their corresponding cell lines are equivalent. Preliminary ELISA experiments showed that the quantification of unesterified HG detected by LM19 in the sodium carbonate fraction followed the same trend in cell cultures as the one observed in the microarray for fruit stages (result not shown). Experiments are in progress to determine if the same occurs with other pectin epitopes.

## 4. Materials and Methods

### 4.1. Plant Material and Callus Development

Strawberry (*Fragaria* × *ananassa* Duch.) plants, cv. ‘Chandler’, micropropagated in N_30_K mineral formulation [65] with MS [32] microelements and vitamins, as described by [34], were used as a source of leaf explants. Fruits at different developmental stages, unripe-green, white, and fully ripe-red, were harvested from ‘Chandler’ plants growing in a greenhouse.

For callus induction, twenty leaf disks from in vitro plants were cultured in Petri dishes containing N_30_K medium supplemented with BA (0, 2.2 and 4.4 µM) and 2,4-D (0, 11.3, 22.6, and 45.2 µM). Fruits, 10–20 fruits per developmental stage, were washed with water and sterilized in a 10% commercial bleach solution with 100 µL of Tween-20 for 15 min. Then, fruits were peeled to remove achenes, sterilized again as previously described, and washed three times with sterile water. Sections of cortical tissue, 0.5 × 0.5 cm, were dissected and cultured in 25 × 150 mm test tubes containing N_30_K medium supplemented with 11.3 µM 2,4-D. In all cases, culture media were solidified with 8 g/L agar and autoclaved for 15 min at 121 °C and 1.05 kg/cm^2^. Explants were cultured in the dark at 25 ± 2 °C and subcultured to fresh medium every two weeks. Once induced, leaf and fruit calli were isolated from the explant and cultured in N_30_K medium supplemented with 11.3 µM 2,4-D, with subculturing at 4 week intervals. Afterwards, a single cell line from each explant type was selected and multiplied for further studies.

The effect of darkness vs. light incubation on the growth of the different cell lines was analyzed. Callus tissues, 0.3 g, were inoculated in test tubes containing 20 mL of N_30_K medium supplemented with 11.3 µM 2,4-D. Cultures were incubated either in darkness or under a 16 h photoperiod at 40 µmol·m^−2^·s^−1^ provided by Sylvania Gro-lux lamps, and the fresh weight was recorded after 4 weeks of culture. Five tubes per treatment and cell line were employed, and the experiment was repeated twice.

### 4.2. Effect of ABA on Anthocyanin Production

Three hundred mg aliquots of each cell line were transferred to test tubes containing N_30_K medium supplemented with ABA at different concentrations (0, 1, 10, and 100 µM) and incubated in light (16 h photoperiod of 40 µmol·m^−2^·s^−1^) for 14 days. Two control treatments consisting of calli cultured in N_30_K medium either supplemented with 11.3 µM 2.4-D or without growth regulators were also employed. After 14 days, the calli were visually inspected for the presence of red coloration, frozen in liquid nitrogen, and stored at −80 °C until used. Five tubes per treatment were employed, and the experiment was repeated twice.

For anthocyanin extraction, frozen callus was pulverized with a mortar and pestle in liquid nitrogen. Then, 0.3 g were incubated in 3 mL MeOH-HCl (99:1, *v/v*) for 4 h at 4 °C in darkness. The homogenate was centrifuged at 10,000× *g* for 15 min, and the absorbance at 515 nm of the supernatant was measured. As pelargonidin 3-glucoside is the main anthocyanin compound in strawberry fruit, anthocyanin concentration was expressed as mg/100 g of callus of pelargonidin 3-glucoside using an extinction coefficient of 17,333 L·mol^−1^·cm^−1^ [66]. Three independent extractions per treatment and cell line were performed.

### 4.3. Cell Wall Extraction

Fruit receptacle and callus from cell cultures grown in N_30_K medium supplemented with 11.3 µM 2,4-D for four weeks were frozen, finely milled to powder under liquid nitrogen, and extracted with PAW (phenol: acetic acid: water, 2:1:1, *w: v: v*) as reported by [67]. Briefly, 10 g of sample powder were extracted with 20 mL of PAW, centrifuged at 4000× *g*, and the pellets were de-starched by aqueous DMSO 90% treatment. The final residue, cell wall extract, was lyophilized prior to fractionation. Three independent extractions per fruit stage and cell line were performed.

### 4.4. Comprehensive Microarray Polymer Profiling (CoMPP)

Cell wall carbohydrate microarrays were performed as previously described [68]. Cell walls were sequentially extracted with sterile water, 0.1 M Na_2_CO_3_, 4 M KOH, and cadoxen (31% 1,2-diaminoethane with 0.78 M cadmium oxide, (*v:v*)) using a tissue lyser (Retsch MM400 mixer mill Retsch GmbH, Haan, Germany). Both alkaline fractions included 0.1% NaBH_4_ freshly added just before use. For the first water-soluble fraction, 10 mg of cell wall extract were homogenized in tissue lyser with 500 µL of water at 30 Hz shaking for 20 min, followed by gentle rocking 1 h at RT. After centrifugation at 2700× *g* for 15 min, supernatants were saved in a fresh tube and stored as water-fraction, while pellets were further extracted with the next solvent following the same extraction steps. Supernatants of each fraction were diluted four times (first dilution 1:1 and five-fold for the following dilutions) and printed as four technical replicates, giving a 16-spots sub-array per sample. All samples were printed simultaneously on the same sheet of nitrocellulose as adjacent arrays. Printing onto nitrocellulose was performed by an ArrayJet Sprint (ArrayJet, Roslin, UK) and quantified as previously described [68]. In brief, the printed nitrocellulose sheets were probed with the primary mAbs diluted (1/10) in phosphate-buffered saline (PBS) containing 5% *w/v* milk powder (MPBS). Secondary anti-rat or anti-mouse antibodies conjugated to alkaline phosphatase (Sigma-Aldrich, St. Louis, Missouri, US) were diluted (1/5000) in MPBS. Primary mAbs used in this study (Table 1 and Appendix A) were from PlantProbes (Leeds, UK) except INRA-RU1 and INRA-RU2, which were kindly provided by M.C. Ralet (Biopolymères Interactions Assemblages, Nantes, France). Developed microarrays were scanned (CanoScan 8800F), converted to TIFFs, and signals were processed by ImaGene 6.0 microarray analysis software (BioDiscovery), as described before [62]. The mean spot signals obtained from four experiments are presented in heat maps in which color intensity was correlated to signal. The highest signal in each dataset was set to 100, and all other values were normalized accordingly.

### 4.5. Statistical Analysis

Data were subjected to analysis of variance (ANOVA) using IBM SPSS Statistics software, version 25. Levene test for homogeneity of variance was performed prior to ANOVA. Tukey and Kruskal–Wallis tests were used for mean separation in the case of homogeneous and non-homogeneous variances, respectively. Normalized values obtained in the carbohydrate microarray experiment were subjected to principal component analysis using the R package FactoMineR. All tests were performed at *p* = 0.05.

## 5. Conclusions

The results obtained in this research suggest that cell cultures obtained from strawberry receptacle at different developmental stages could be a useful model system to achieve a better understanding of the ripening process in strawberry. Metabolic pathways involved in anthocyanin production or hormonal regulation of ripening could be investigated using this system. Despite the differences observed in cell wall composition in fruits and callus cultures, some aspects of cell wall disassembly during fruit development, e.g., pectin remodeling, could also be addressed using this system.

## Figures and Tables

**Figure 1 plants-09-00805-f001:**
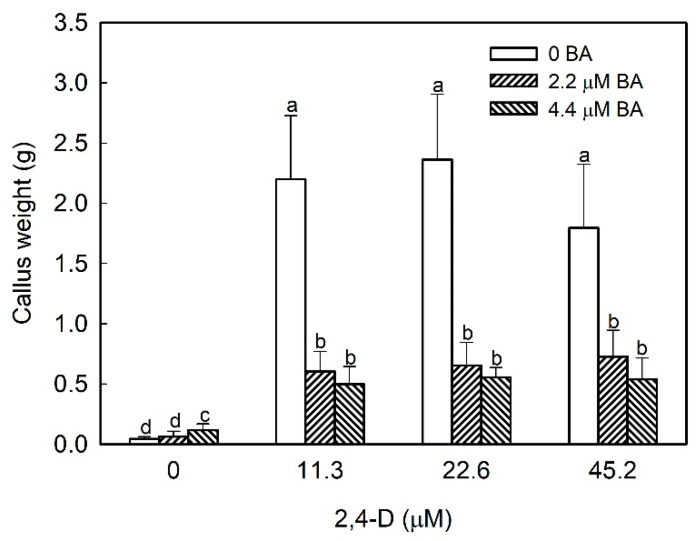
Effect of hormonal balance in callus development from strawberry leaf explants cultured in N_30_K medium. Data were taken after 8 weeks of culture in the dark. Means with different letters indicate significant differences by Kruskal–Wallis test at *p* = 0.05.

**Figure 2 plants-09-00805-f002:**
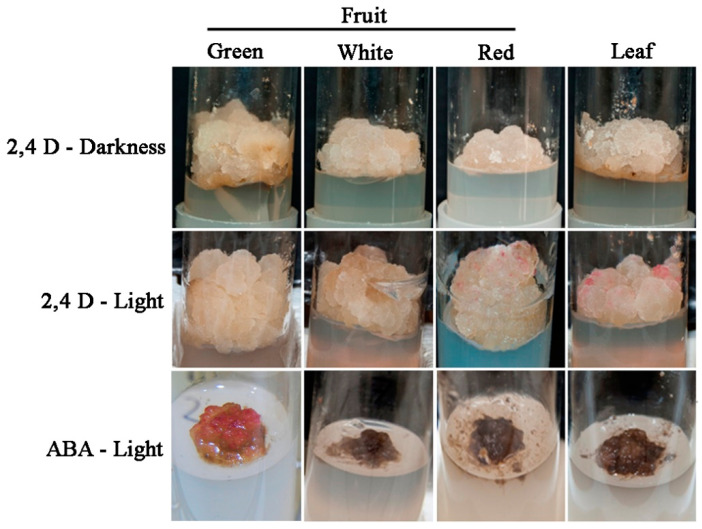
Aspect of calli obtained from leaf and fruit at different developmental stages growing in N_30_K medium supplemented with 11.3 µM 2,4-Dichlorophenoxyacetic acid (2,4-D) in darkness (2,4 D-darkness), the same medium under a 16 h photoperiod at 40 µmol·m^−2^·s^−1^ (2,4-D light) or N_30_K medium with 1 µM abscisic acid (ABA) under light conditions (ABA-light). Pictures were taken after 4 weeks of culture in the presence of 2,4-D and 2 weeks in the case of ABA.

**Figure 3 plants-09-00805-f003:**
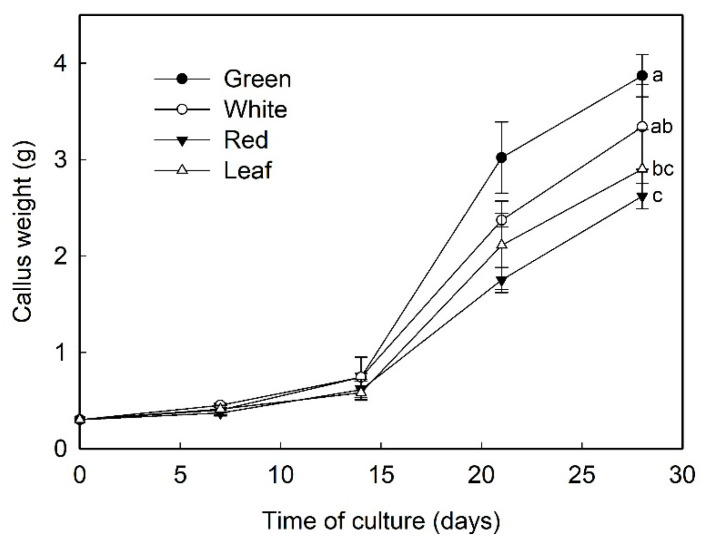
Growth curve of the different callus lines obtained from leaf and fruit receptacle at green, white, and red stages. Data represent mean ± SD. At 28 days of culture, means with different letters indicate significant differences by Tukey test at *p* = 0.05.

**Figure 4 plants-09-00805-f004:**
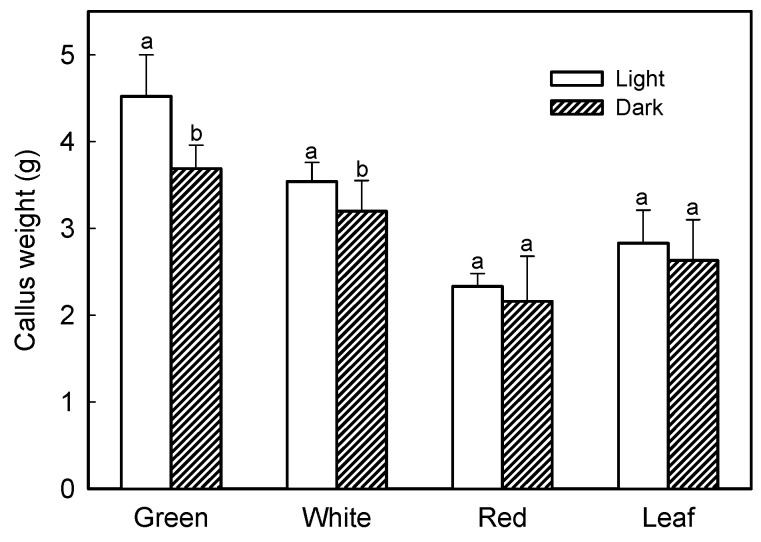
Effect of light incubation in the growth of calli obtained from strawberry leaf and fruits at different developmental stages. Within each callus line, columns with different letters indicate significant differences by Student’s *t*-test at *p* = 0.05.

**Figure 5 plants-09-00805-f005:**
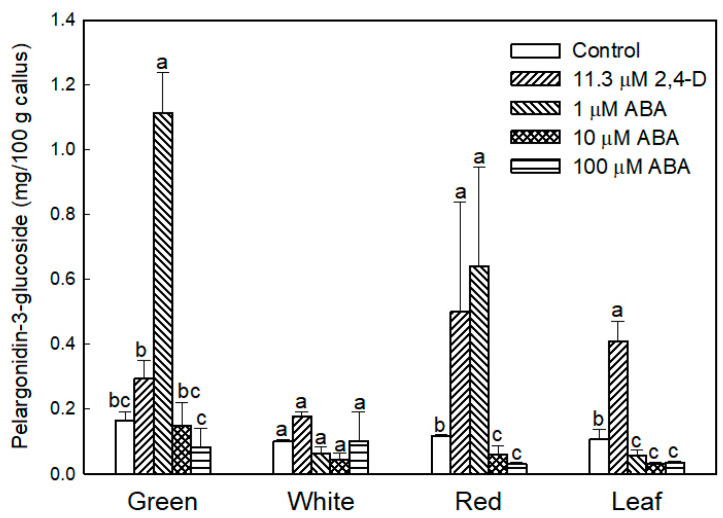
Anthocyanin production in calli obtained from strawberry leaf and fruits at different developmental stages after culturing for 2 weeks in N_30_K medium with 2,4-D or ABA in the presence of light. Control: callus incubated in the absence of growth regulators. Within each callus line, columns with different letters indicate significant differences by Kruskal–Wallis test at *p* = 0.05.

**Figure 6 plants-09-00805-f006:**
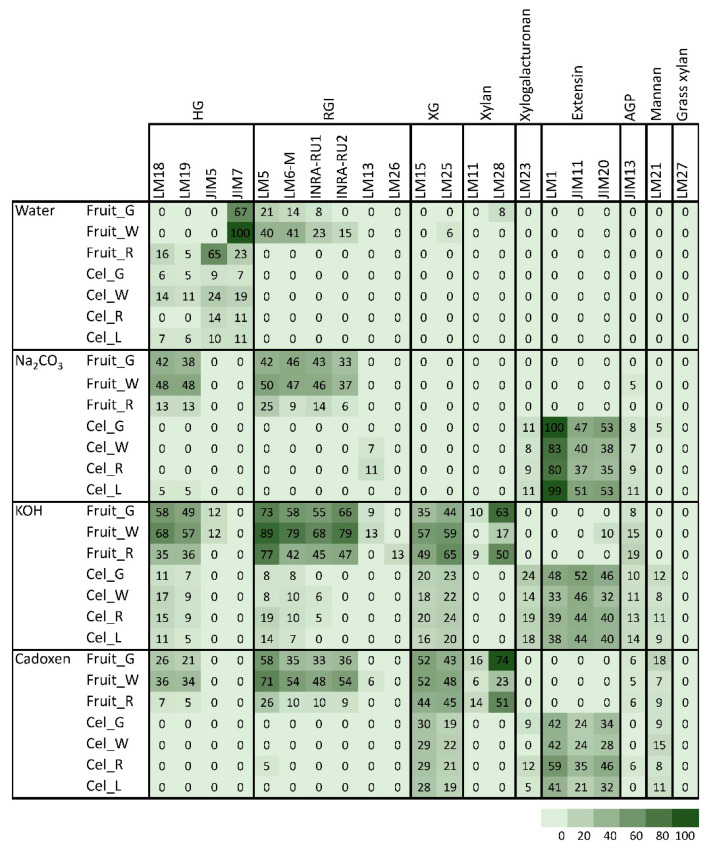
Heat map showing the relative abundance of cell wall epitopes recognized by different mAbs in cell wall fractions extracted from strawberry fruits and cell cultures obtained from leaf and fruit receptacle at different developmental stages. A value of 100 was assigned to the highest mean spot signal, and all other signals were adjusted accordingly. Fruit_G: unripe-green fruit; Fruit_W: white fruit; Fruit_R: mature-red fruit; Cel_G: callus from green fruit; Cel_W: callus from white fruit; Cel_R: callus from mature-red fruit; Cel_L: callus from leaf.

**Figure 7 plants-09-00805-f007:**
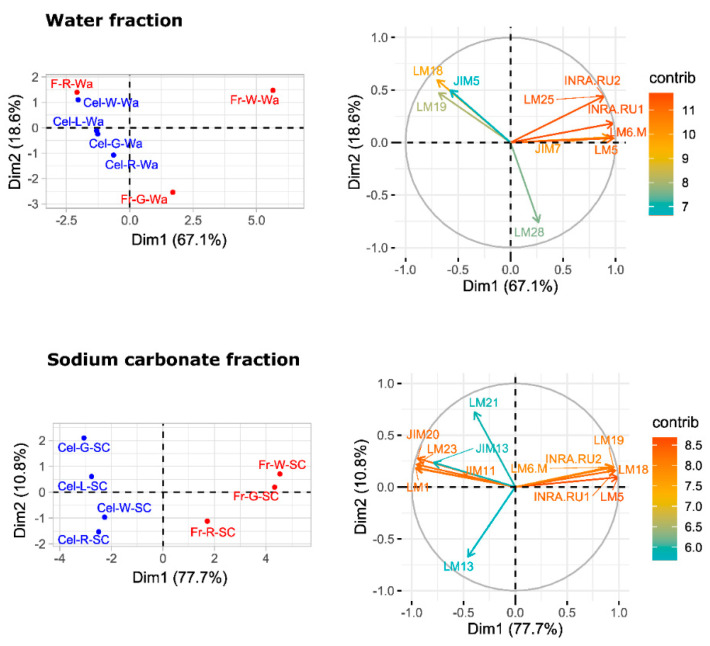
PCA analysis of carbohydrate microarray data from pectin enriched fractions, water, and sodium carbonate fractions. The factor score plots are shown in the **left** and the variable plots in the **right**. The color scale in the variable plots represents the average contribution of the variable to the variation explained by the two principal components. Fr: fruit; Cel: callus culture; G: green stage; W: white stage; R: red stage; Wa: water fraction; SC: sodium carbonate fraction.

**Figure 8 plants-09-00805-f008:**
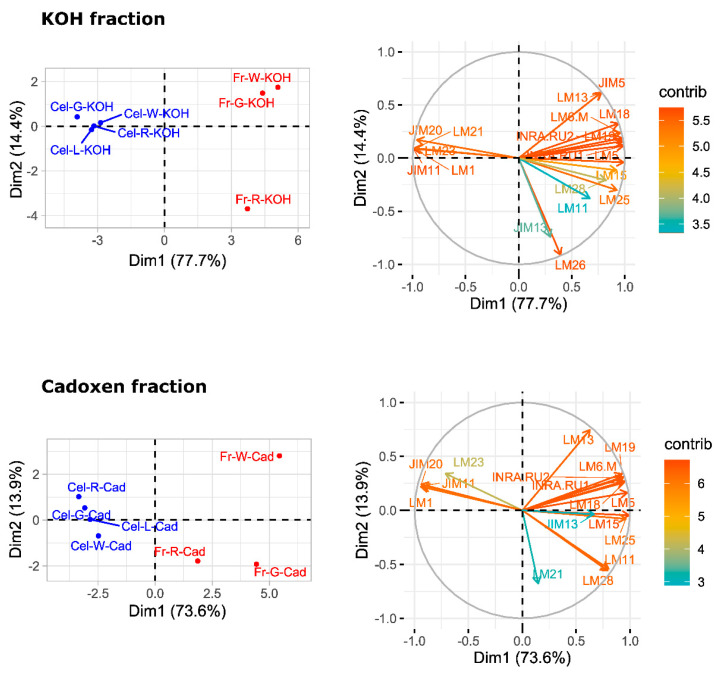
PCA analysis of carbohydrate microarray data from matrix glycan fractions, xyloglucans (KOH), and cadoxen fractions. The factor score plots are shown in the **left** and the variable plots in the **right**. The color scale in the variable plots represents the average contribution of the variable to the variation explained by the two principal components. Fr: fruit; Cel: callus culture; G: green stage; W: white stage; R: red stage; KOH: 4M KOH fraction; Cad: cadoxen fraction.

**Figure 9 plants-09-00805-f009:**
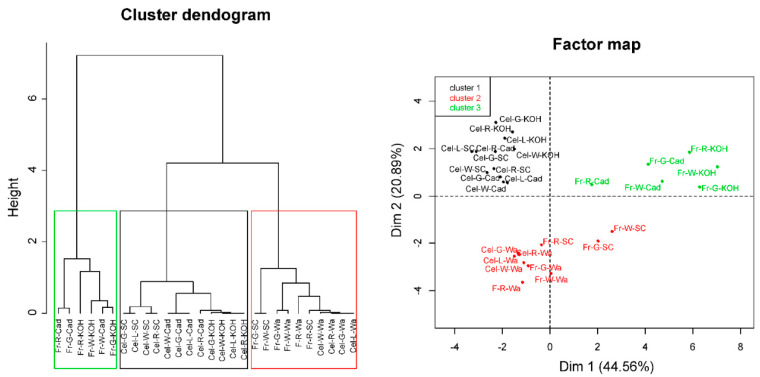
Hierarchical clustering of carbohydrate microarray data. Cluster dendogram based on Euclidean distances (**left**) and the corresponding individual factor map (**right**) of the different cell wall samples. Fr: fruit; Cel: callus culture; G: green stage; W: white stage; R: red stage; Wa: water fraction; SC: sodium carbonate fraction; KOH: 4M KOH fraction, Cad: cadoxen fraction.

**Table 1 plants-09-00805-t001:** List of monoclonal antibodies used in the carbohydrate microarray and the epitopes recognized by each mAb.

Antibody	Cell Wall Epitope
LM18, LM19, JIM5	Partially Me- homogalacturonan (HG)/no ester
JIM7	Partially Me-HG
LM5	(1→4)-β-d-galactan
LM6-M	(1→5)-α-l-arabinan
INRA-RU1, INRA-RU2	[→2)-α-l-rhamnose-(1→4)-α-d-galacturonic acid-(1→]_7_
LM13	Linearized (1→5)-α-l-arabinan
LM26	Branched galactan
LM15	Xyloglucan (XXXG motif)
LM25	XXXG/galactosylated xyloglucan
LM11	(1→4)-β-d-xylan/arabinoxylan
LM23	Non-acetylated xylosyl
LM28	Glucuronoxylan
LM1, JIM11, JIM20	Extensin
JIM13	Arabinogalactan-protein (AGP) glycan
LM21	Heteromannan
LM27	Grass heteroxylan

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
