# Peer review of "Exploring the Use of Fruit Callus Culture as a Model System to Study Color Development and Cell Wall Remodeling during Strawberry Fruit Ripening"

_plants, 2020, doi:10.3390/plants9070805_

Round 1

Reviewer 1 Report

The manuscript deals with an interesting   opportunity to use plant tissue culture as a model system to study different aspects of strawberry fruit development. The use of plant cell culture as model system for elucidate different plant responses and complex process as fruit cell senescence was just experimented (see the old works of RJ Romani et al in 1980-90), but not in strawberry fruit, so this work is enough original to be published, but the discussion and the proposed conclusions are not completely sustained by the results so major revisions must be done before acceptance and publication.

RESULTS

  • As regards the comparison between different mineral media in promoting growth and accumulation of anthocyanin, the experiment is not well set up, in fact, the effects of two mineral media cannot be compared if they are combined with different plant growth regulators, LS and NK should be tested all other factors being equal. Therefore the LS/ NK comparison must be eliminated from the manuscript and fig 4 must be modified ( adding only data on light effect ) or eliminated by introducing in the text the average values obtained on the medium NK with 2,4, D in the light and in the dark to evaluate the effect of the light
  • In literature is widely reported the role of citokinins in anthocyanins promotion, so why the author did not report any data on the effect of the tested citokinins on callus pigmentation but they referrer only to the effect on callus growth?

DISCUSSION

  • Line 280: …..although callus proliferation was reduced at high auxin concentrations……..Callus growth was not statistical different on media with the three increasing concentration of 2,4D tested (fig 1)
  • Line 295 ….Light exerted a minor effect on the proliferation rate of callus cultures; however, it induced the accumulation of red pigmentation in leaf and ripe-fruit lines……In fig 4 significant differences were reported in callus growth under dark and light condition, so you cannot write” minor effect”. On the contrary, in fig 5 no evident effect on pigmentation was reported and the only evidence is a light red pigmentation of calli in the fig 2. More experimental data must be added to sustain the discussed effect of light
  • In various lines you report the effect of light on pigmentation, but data on anthociyanins in dark or light condition are not clear reported and must be deduced… . Please introduce a table or a figure, similar to fig 5, with data of anthocyanins in the dark and light in the four type of callus on the same medium.

Author Response

  • As regards the comparison between different mineral media in promoting growth and accumulation of anthocyanin, the experiment is......

Fig.4 has been modified as requested and the results related to LS medium have been deleted

  • In literature is widely reported the role of citokinins in anthocyanins promotion, so why the author did not report any data on the effect of the tested citokinins on callus pigmentation but they referrer only to the effect on callus growth?

We found that citokinins were not needed neither during the callus induction phase nor callus proliferation phase (see results of first experiment). Then, the role of citokinins in callus pigmentation was not evaluated. On the other hand, ABA has been reported as the main hormone in strawberry fruit development, and for this reason we focused in the role of ABA in color development in vitro

  • Line 280: …..although callus proliferation was reduced at high auxin concentrations……..Callus growth was not statistical different on media with the three increasing concentration of 2,4D tested (fig 1)

This comment has been deleted

  • Line 295 ….Light exerted a minor effect on the proliferation rate of callus cultures; however, it induced the accumulation of red pigmentation in leaf and ripe-fruit lines……In fig 4 significant differences were reported in callus growth under dark and light condition, so you cannot write” minor effect”.

This sentence has been modified as follow: "Light slightly increased the proliferation rate of green and white callus cultures…"

  • On the contrary, in fig 5 no evident effect on pigmentation was reported and the only evidence is a light red pigmentation of calli in the fig 2. More experimental data must be added to sustain the discussed effect of light

No red pigmentation was observed in any callus cultures when they were grown in darkness. When calli were transferred to light, some sector of cells developed red color. We have included mean values on anthocyanin content in calli cultured in darkness vs. light, as requested (line 141, new version)

  • In various lines you report the effect of light on pigmentation, but data on anthociyanins in dark or light condition are not clear reported and must be deduced… . Please introduce a table or a figure, similar to fig 5, with data of anthocyanins in the dark and light in the four type of callus on the same medium.

Data on anthocyanin production in calli cultured in light have been included as requested (line 141, new version).  

Reviewer 2 Report

Dear Editor,

The manuscript (plants-845196) is targeted to study hormonal regulation of  color development and cell wall remodeling process during strawberry fruit ripening in callus induced from fruit and leaves. It was shown that cell cultures obtained from strawberry is a good model to study a ripening process in strawberry but it has some limitation when studying the process of cell wall remodeling leading to fruit softening due to differences observed in cell wall composition in fruits and callus cultures. The article is well-written and discussed. The title properly reflects the subject of the paper. The introduction summarizes resent research related to the topic. Manuscript structure is clear. Figures and Supplementary Table 1  are clear.

My only remark relates to the abbreviations of growth regulators, the names of the plant media that in my opinion should be explained when they are used for the first time. The abbreviations like ABA, BA, 2,4-D, MS, LS are commonly known however Plants does not publish only publications related to in vitro cultures and clarifying the abbreviations can make it easier to read  the article for a wider audience. I think the manuscript is suitable for publication with minor revision.

Similarly, Plant cell monoclonal Antibody e.g., JIM5, JIM7 etc. are explained only in supplementary materials.

Author Response

  • My only remark relates to the abbreviations of growth regulators, the names of the plant media that in my opinion should be explained when they are used for the first time. The abbreviations like ABA, BA, 2,4-D, MS, LS are commonly known however Plants does not publish only publications related to in vitro cultures and clarifying the abbreviations can make it easier to read  the article for a wider audience. I think the manuscript is suitable for publication with minor revision.

Full name of these compounds and culture media have been included as requested

  • Similarly, Plant cell monoclonal Antibody e.g., JIM5, JIM7 etc. are explained only in supplementary materials.

The list of antibodies used in the microarray experiment has been included in the text (Table 1)

Reviewer 3 Report

The paper presents result on the effects of fitoregulators on growth and differentiation of cell culture, calli, from strawberry organs and tissues. The experiments are properly set and the results appears interesting and deserve to be published. Nevertheless some points should be modified, clarified, and discussed.

-Line 62-63. Usually Calli are not a homogeneous material especially when becomes of consistent

dimension depending on the different supplying of nutrients, etc... This should be discussed and clarified.

-The results on the cell wall compositions are interesting but the fact that cell wall is very different from the naturally differentiated tissues is evident so the rationale of this analysis should better introduced and discussed. In addition it is not clear if the calli obtained to study cell wall are obtained using 2,4, D.

-Callus growth rate is not well described: the amount of initial mass (weight) should be introduced

-2,4,D and BA are not hormones. In many points of the manuscript they are defined as hormones.

-The light quality should be specified.

Author Response

  • Line 62-63. Usually Calli are not a homogeneous material especially when becomes of consistent dimension depending on the different supplying of nutrients, etc... This should be discussed and clarified.

Some heterogeneity among callus sectors can be found, but as indicated in the text, this system is simpler and easier to handle that intact organs.

  • The results on the cell wall compositions are interesting but the fact that cell wall is very different from the naturally differentiated tissues is evident so the rationale of this analysis should better introduced and discussed. In addition it is not clear if the calli obtained to study cell wall are obtained using 2,4, D.

The fact that cell walls from in vitro cells are very different from natural tissue is not so evident, mainly because there are very few studies comparing cell walls from both kind of samples, as it is indicated in the introduction.

Cell walls were extracted from calli cultured in 2,4-D (line 434, new version)

  • Callus growth rate is not well described: the amount of initial mass (weight) should be introduced

Initial callus weight was 0.3 g as indicated in the old version (line 414, new version)

  • 2,4,D and BA are not hormones. In many points of the manuscript they are defined as hormones.

We disagree with this comment. 2,4-D and BA are very well-known plant hormones. Please, see the landmark book Plant Physiology and Development by Taiz et al.

  • The light quality should be specified.

It has been included in the new version (line 416)

Round 2

Reviewer 1 Report

I agree with the author responses, so now, in the present form , the manuscript can be published